# The Multifaceted S100B Protein: A Role in Obesity and Diabetes?

**DOI:** 10.3390/ijms25020776

**Published:** 2024-01-08

**Authors:** Fabrizio Michetti, Gabriele Di Sante, Maria Elisabetta Clementi, Federica Valeriani, Martina Mandarano, Francesco Ria, Rosa Di Liddo, Mario Rende, Vincenzo Romano Spica

**Affiliations:** 1Istituto di Scienze e Tecnologie Chimiche “Giulio Natta” SCITEC-CNR, L.go F. Vito 1, 00168 Rome, Italy; elisabetta.clementi@scitec.cnr.it; 2Department of Neuroscience, Catholic University of the Sacred Heart, L.go F. Vito 1, 00168 Rome, Italy; 3Department of Medicine, LUM University, 70010 Casamassima, Italy; 4Genes, Via Venti Settembre 118, 00187 Roma, Italy; 5Department of Medicine and Surgery, Section of Human, Clinical and Forensic Anatomy, University of Perugia, 06132 Perugia, Italy; gabriele.disante@unipg.it (G.D.S.); mario.rende@unipg.it (M.R.); 6Department of Movement, Human and Health Sciences, University of Rome “Foro Italico”, 00135 Rome, Italy; federica.valeriani@uniroma4.it (F.V.); vincenzo.romanospica@uniroma4.it (V.R.S.); 7Department of Medicine and Surgery, Section of Anatomic Pathology and Histology, Medical School, University of Perugia, 06132 Perugia, Italy; martina.mandarano@unipg.it; 8Department of Translational Medicine and Surgery, Section of General Pathology, Catholic University of the Sacred Heart, 00168 Rome, Italy; francesco.ria@unicatt.it; 9Department of Pharmaceutical and Pharmacological Sciences, University of Padova, 35131 Padova, Italy; rosa.diliddo@unipd.it

**Keywords:** S100B protein, adipose tissue, obesity, diabetes

## Abstract

The S100B protein is abundant in the nervous system, mainly in astrocytes, and is also present in other districts. Among these, the adipose tissue is a site of concentration for the protein. In the light of consistent research showing some associations between S100B and adipose tissue in the context of obesity, metabolic disorders, and diabetes, this review tunes the possible role of S100B in the pathogenic processes of these disorders, which are known to involve the adipose tissue. The reported data suggest a role for adipose S100B in obesity/diabetes processes, thus putatively re-proposing the role played by astrocytic S100B in neuroinflammatory/neurodegenerative processes.

## 1. Introduction

The S100B protein, which is known to be abundant in the nervous system, where it is concentrated in astrocytes, is also present in other districts, and expressed by other cell types, such as adipocytes, where, intriguingly, it is especially concentrated [1,2,3]. While the possible role(s) of S100B in the nervous system has been widely addressed, the relationship between S100B and adipose tissue has not been extensively studied, and the interaction or involvement of the protein in standard physiological processes of this tissue is not well-established. However, consistent research has shown interesting associations between S100B and adipose tissue in the context of obesity, metabolic disorders, and diabetes.

Obesity is defined as abnormal or excessive fat accumulation that presents a health risk, characterized by a body mass index (BMI) above 30 [4]. This condition is often associated with metabolic disorders, such as hyperglycemia, dyslipidemia, and hypertension. It is also characterized by chronic low-grade inflammation and insulin resistance, collectively known as metabolic syndrome [5]. Together with overweight subjects (BMI > 25), the number of obese subjects is continuously growing. According to the World Health Organization, the prevalence of overweight or obese children and adolescents aged 5–19 years from 1975 to 2016 increased from 4% to 18%. Globally, obesity is regarded as a pathological condition, and its pathogenesis includes an interplay between environmental and genetic factors, involving primarily, although not exclusively, the adipose tissue. In this context, neurotransmitters and hormones also appear to affect food intake, fat metabolism, and energy balance [6].

The term diabetes refers to different disorders of metabolism having multiple etiology but characterized by disturbances of carbohydrate, fat, and protein metabolism associated with chronic hyperglycemia resulting from defects in action or secretion of insulin [7,8,9]. The global diabetes prevalence in 2019 was estimated to be 9.3% (463 million people), rising to 10.2% (578 million) by 2030 and 10.9% (700 million) by 2045 [10]. Obesity is regarded to constitute an important risk factor for developing diabetes [11,12], especially type 2 diabetes, which is considered to be the most frequent form of this disease, characterized by resistance to insulin, so the term “diabesity” has also been used to include both syndromes [13]. This condition appears to be strongly associated with the pathophysiological changes in the adipose tissue, where chronic inflammation, dysregulated glucose homeostasis, and impaired adipogenesis lead to the accumulation of ectopic fat, and insulin resistance may be observed [14,15].

## 2. The S100B Protein

The term S100B refers to a protein identified in the mid-sixties of the last century from brain extracts where, using procedures available at that time (essentially chromatography and starch gel electrophoresis), it was originally regarded to be specific for that tissue. The isolated protein was also shown to be soluble in a 100% saturated solution with ammonium sulfate. This characteristic was at the basis of its denomination, which originally was simply S100 protein. At present, the S100 protein family comprises more than 20 calcium-binding proteins, mostly homodimers, exhibiting structural similarities, located in different tissues, where they modulate the activity of many targets, sometimes peculiar to the cell type where they are located. They constitute the largest subgroup within the EF-hand protein superfamily, which is characterized by a calcium-binding loop forming a conserved pentagonal arrangement around the calcium ion (EF-hand motif). Interestingly, some members of the S100 protein family may bind zinc and/or copper, thus suggesting the possibility that these metals might participate in the regulation of their biological activity. S100B is an acidic homodimer (2 beta subunits) of 9–14 kDa per monomer and constitutes the bulk of the protein fraction, which was isolated originally from brain extracts, and during approximately two decades, as above indicated, S100B, which at that time was named merely S100, has been regarded to be specific for this tissue. Interestingly, the amino acidic composition and conformation of S100B, as for other proteins of the S100 family, is highly conserved in different species, suggesting that it may have a crucially preserved biological role(s). Interestingly, in this respect, an S100-like protein has even been immunologically detected in planarians. In the nervous system, S100B is concentrated in astrocytes and is also expressed in other glial cell types, such as oligodendrocytes, Schwann cells, ependymal cells, retinal Müller cells, and enteric glial cells, and has been located even in specific neuronal subpopulations in the brainstem and in some ganglionic peripheral cells. However, it has also been demonstrated that the protein is not restricted to nervous tissue. After the original finding in human skin, where the protein was located in melanocytes and Langerhans cells, S100B has also been detected in definite non-neural cell types: chondrocytes, dendritic cells of lymphoid organs, some lymphocyte cell types, adrenal medulla satellite cells, skeletal muscle satellite cells, tubular kidney cells, non-nervous structures of the eye, Leydig cells, and, in particular, adipocytes, which intriguingly constitute a site of concentration for the protein comparable to astrocytes. While neural S100B has been extensively studied, its properties in non-neural locations have received poor attention, although they would reasonably deserve analogous consideration during physiological and/or pathological conditions [1,2,3]. Indeed, the cell distribution of this protein does not offer conclusive clues to its functional role(s). In general, S100B, as a calcium-sensor protein, appears to regulate a variety of intracellular activities, transferring signals from second messengers and interacting with different molecules in different cell types. But, at present, the different functions attributed to the protein (e.g., cell proliferation, survival and differentiation, participation in the regulation of cellular calcium homeostasis and enzyme activities, and interaction with cytoskeleton) do not appear to delineate a clear univocal intracellular role for S100B. In contrast, after the first demonstration of S100B in the extracellular compartment in the late seventies of the last century, when elevated levels of the protein were detected in the cerebrospinal fluid of multiple sclerosis patients in the acute phase, whereas lower levels were found in the stationary phase of the disease [16], growing evidence indicates an increasingly clearer role for S100B when secreted in the extracellular compartment. Extracellular S100B is regarded to interact with target cells mainly, but not only, through the multi-ligand transmembrane Receptor for Advanced Glycation End-products (RAGE) initiating intracellular signaling cascades, which may result in physiological regulation at low nanomolar concentrations (“Jekyll side”), or various pathological conditions, acting as a Danger/Damage Associated Molecular Pattern (DAMP) protein, at higher micromolar concentrations (“Hyde side”) [2,3,17]. Interestingly, some characteristics of S100B, such as its binding with RAGE, its non-canonical secretion modality that bypasses the classical Golgi route, and its ability to stimulate microglial migration, are shared with DAMPs [18,19].

## 3. The S100B Protein in Adipose Tissue

Adipose tissue is especially rich in the S100B protein and its mRNA [20,21].

Traditionally, adipose tissue is considered a homogeneous entity with a primary function as a storage site for excess energy. However, this tissue is regarded to consist of various depots located throughout the body with distinct characteristics. White Adipose Tissue (WAT) is the predominant type, mainly responsible for energy storage, and it is associated with the production of pro-inflammatory adipokines and can contribute to chronic low-grade inflammation and metabolic dysregulation. Brown Adipose Tissue (BAT), primarily involved in energy expenditure and thermogenesis, shows a higher density of mitochondria, is rich in blood vessels and nerves, and has been suggested to be implicated in immune modulation [22]. Subcutaneous Adipose Tissue (SAT), although it is the most abundant depot in the body, has a lower association with obesity-related complications, whereas Visceral Adipose Tissue (VAT), which surrounds internal organs, is metabolically more active and releases a higher amount of pro-inflammatory adipokines compared to SAT. VAT is strongly associated with metabolic syndrome, insulin resistance, and an increased risk of cardiovascular diseases [23,24]. Adipose tissue produces and releases various signaling molecules, collectively known as adipokines, such as adiponectin, leptin, and resistin [25,26,27,28]. These adipokines can have pro-inflammatory or anti-inflammatory effects, depending on their types and concentrations, and can impact immune function and inflammation. Interestingly, these different components of the adipose tissue are also regarded to form, at least in mice and humans, a large unitary structure, fulfilling the requirements to be considered a true adipose organ [29]. The immune system is closely intertwined with adipose tissue, and immune cells are present within the adipose tissue environment and can secrete cytokines and chemokines, influencing local inflammation and immune responses [30,31,32]. In obesity, immune cells infiltrate into adipose tissue, leading to an altered immune response and increased inflammation [33,34,35,36]. It has also been demonstrated that specific adipose tissues can represent the source of immune response in multifactorial immune-related disorders, as demonstrated by epicardial adipose tissue and myocardial infarction [30,37]. These data corroborate the idea that the interaction between adipose tissue and the immune system is complex and bidirectional and can contribute to the development of metabolic disorders, insulin resistance, and other obesity-related complications. Understanding the mechanisms underlying this relationship is essential for developing strategies to manage obesity-related disorders. To this purpose, identifying specific molecular targets and pathways that can be modulated to regulate inflammation and improve metabolic health in individuals with obesity may be especially useful.

The adipose tissue constitutes a site of concentration for S100B comparable to the nervous tissue and is also regarded to release the protein [38]. In particular, the S100B protein was found in white adipocytes and warm acclimatized pauci- or unilocular inactive brown adipocytes [39], whereas a role for the protein in adaptive adipocyte-dependent thermogenetic mechanisms has also been hypothesized [40]. The secretion of adipose S100B has also been proposed to stimulate thermogenesis via activation of sympathetic innervation [41].

In experimental animals and tissues, S100B is released from adipose tissue under hormonal control since the eighties of last century (Table 1). In rat epidydimal fat pads and isolated adipocytes in vitro, S100B was shown to be released under epinephrine or adrenocorticotropin (ACTH) stimulation [42]. The intracellular behavior of S100B during lipolysis processes was also followed by immunoelectron microscopy in epidydimal fat pads from Wistar rats, where it was found in numerous macrovesicles frequently fusing with the plasma membrane and opening into the interstitium [43]. Also in vivo, the amount of S100B was significantly reduced in the epididymal adipose tissue of Wistar rats after serial (inducing reduction of more than 50%) or a single injection of catecholamines [44]. The release of S100B induced by epinephrine, ACTH, or isoproterenol in rat epidydimal fat pads was reduced by approximately 50% in the presence of insulin, possibly by a cyclic AMP-mediated mechanism; similar results were obtained in fat pads of insulin-injected rats. In contrast, in the fat pads obtained from diabetic or long-term starved rats, the S100B protein release was greatly enhanced, showing several-fold higher levels of basal release in the absence of hormones, and S100B protein contents in the epididymal adipose tissues of these rats were significantly lower than those of the control: these results suggested that the S100B protein content in adipocytes may be regulated by insulin as well as lipolytic hormones [45]. The final aim of this review is a possible stimulation of research addressing the disregarded topic of adipose S100B.

In addition, the overexpression of the S100B receptor RAGE in 3T3-L1 preadipocytes using adenoviral gene transfer accelerated adipocyte hypertrophy, associated with attenuated insulin-stimulated glucose uptake, and insulin-stimulated signaling, whereas inhibitions of RAGE by small interfering RNA significantly decreased adipocyte hypertrophy. Interestingly, the knockdown of S100B, associated with the knockdown of high mobility group box-1, both of which are DAMPs and RAGE ligands, canceled RAGE-induced adipocyte hypertrophy, implicating a fundamental role of the interaction of RAGE with ligands in the phenomenon [46].

Noticeable reviews addressing the possible role(s) of S100 proteins, including S100B, in adipose tissue, have also been published at different times [38,47]. This review tunes the possible role of S100B in pathogenic processes of obesity/diabetes interpreting together recent and less recent data, the latter having been probably disregarded and received inadequate attention in recent times.

## 4. S100B Protein in Obesity/Diabetes and Related Conditions

A preliminary consideration necessarily points out that most of the reported data indicate more of a correlation than a causation between S100B and obesity/diabetes. Having the above consideration in mind, whereas S100B levels in different biological fluids (cerebrospinal fluid, blood, urine, amniotic fluid, and saliva) are considered a reliable biomarker to detect and/or monitor disorders of the nervous system (and also melanoma) [48], data regarding the use of S100B as a biomarker in human biological fluids, namely, blood, concerning glucose metabolism and body mass have also been obtained in the first two decades of this century (Table 2); high levels of S100B in the blood have been shown to be elevated in individuals with obesity in different conditions and have also been shown to be positively correlated with abdominal obesity, serum levels of triglyceride, and insulin resistance [49,50,51]. Interesting observations concerning human blood levels of S100B and their correlations with conditions of the adipose tissue or glucose metabolism have been shown. In this respect, blood levels of the protein have been shown to decrease in chronic starvation, whereas they are normalized with weight gain [52]. Blood S100B levels have also been shown to be related to insulin release. In particular, during oral glucose tolerance tests, they correlated inversely with the insulin response [53]. In addition, subjects with metabolic syndrome have been shown to have significantly higher blood levels of S100B than healthy subjects [50]. In schizophrenic patients, where blood S100B levels are known to be elevated, they have been shown to be related to insulin resistance and, also, to visceral obesity [51,54].

In addition to the above indicated data in human blood, converging evidence, obtained in tissues from experimental models, potentially attributes a putative role to S100B in obesity and diabetes pathogenic processes. In this respect, S100B might contribute to low-grade inflammation in adipose tissue and promote the development of obesity-related complications.

Indirect evidence regarding S100B points to the involvement of RAGE, the primary receptor of S100B, in the pathophysiology of adipose tissue, essentially indicating that RAGE-mediated adipose tissue inflammation and insulin-signaling are potentially important mechanisms that contribute to the development of obesity-associated insulin resistance [55,56,57]. As above indicated, overexpression of RAGE in 3T3-L1 preadipocytes has been shown to induce adipocyte hypertrophy, together with attenuated insulin-stimulated glucose uptake, whereas inhibitions of RAGE significantly decreased adipocyte hypertrophy. In addition, Toll-like receptor (Tlr2) mRNA, a receptor that is believed to be involved in inflammatory processes of different tissues, including the adipose tissue [58,59], was upregulated by RAGE overexpression, and inhibition of Tlr2 abrogated RAGE-mediated adipocyte hypertrophy. Finally, RAGE knockout mice, which are unable to synthesize RAGE, exhibited significantly reduced body weight, epididymal fat weight, epididymal adipocyte size, and higher insulin sensitivity as compared with wild-type mice, whereas RAGE deficiency was also associated with suppression of Tlr2 mRNA expression in adipose tissues [46].

Other data are regarded directly as S100B (Table 2). Both S100B expression in white adipose tissue and S100B plasma levels were significantly increased in diet-induced obese mice [60], and tissue S100B increase was reversed following weight loss, in indirect accordance with results obtained in human sera [49]. Taken together, the bulk of data may suggest that S100B might play a role as an inflammatory adipokine in the interaction between adipocytes and macrophages to establish a vicious paracrine loop (Figure 1), possibly as a part of the adipocyte/macrophage cross-talk, which has been described in obesity [61], thus reflecting the putative role played by astrocytic S100B, via microglial cells, in neuroinflammatory/neurodegenerative disorders [18]. Indeed, inflammatory processes involving DAMPs, as S100B is considered [2,3], are widely regarded to be active in obesity/diabetes [62]. In this respect, recombinant S100B was shown to upregulate tumor necrosis factor-α (TNF-α) and M1 (mainly involved in proinflammatory responses) proinflammatory markers in murine RAW264.7 macrophages. In turn, TNF-α stimulated S100B secretion from 3T3 L1 adipocytes, whereas conditioned media from these cells stimulated TNF-α secretion from macrophages, and macrophage conditioned media increased S100B secretion from adipocytes [63]. This may be especially relevant in light of the key role that macrophages are believed to play in the adipose tissue as mediators of inflammation and insulin resistance during processes leading to obesity [64].
ijms-25-00776-t002_Table 2Table 2Behavior of S100B in obesity, diabetes, and related conditions.Human Blood or Experimental ModelConditionS100B BehaviorReferenceHuman bloodOverweight, visceral obesity, insulin resistanceElevated levelsSteiner et al., 2010 [49]Human bloodMetabolic syndromeElevated levelsKheirouri et al., 2018 [50]Human bloodHigh or low body mass indexElevated or reducedSteiner et al., 2010 [51]Human bloodChronic starvationReduced levels normalizing with weight gainHoltkamp et al., 2008 [52]Human bloodOral glucose tolerance testInverse correlation with insulin secretionSteiner et al., 2014 [53]Human bloodInsulin resistanceElevated levelsSteiner et al., 2010 [51]Mouse blood and white adipose tissueDiet-induced obesityElevated levels normalizing with weight lossBuckman et al., 2014 [60]3T3-L1 adipocytesStimulation by TNF-a or murine RAW264.7 macrophage-conditioned mediaIncreased secretionFujiya et al., 2014 [63]Murine RAW264.7 macrophagesWild-typeInduction of upregulation of TNF-a and M1 proinflammatory markersFujiya et al., 2014 [63]Diabetic OLETF ratWild-typeExpression with RAGE in Islet cellsLee et al., 2010 [65]INS-1 cells and rat, pig, and human islets (b cells)Wild-typeInduction of apoptosisLee et al., 2010 [65]S100B knockout miceUntreatedInduction of resistance streptozotocin-derived diabetesMohammadzadeh et al., 2018 [66]


It may also be relevant that S100B and its receptor RAGE were found to be expressed in islet cells of 28-week-old diabetic OLETF rats, a recognized model of type 2 diabetes, and that S100B induced apoptotic cell death of pancreatic β-cells via oxidative stress. These data have been regarded to indicate that the S100B/RAGE interaction participates in the progressive β-cell loss in type 2 diabetes [65]. S100B also induced reactive oxygen species-dependent and RAGE-dependent apoptosis in pancreatic β-cells derived from wild-type mice. In addition, S100B knockout mice, unable to synthesize S100B, are resistant to diabetes induced by streptozotocin, exhibiting enhanced insulin sensitivity, glucose tolerance, prevention of β cell destruction, and lower urine volume, food, and water intake compared to wild-type mice. Thus, S100B has been proposed as a potential therapeutic target for diabetic processes, although the peculiarities of the experimental model used should be taken into account. In particular, the consideration that streptozotocin merely kills beta cells might restrict the importance of the observation [66].

Another participant in obesity/diabetes (diabesity) processes, putatively involving S100B, is the gut microbiome [67]. This consists of millions of microorganisms present in the human intestinal apparatus, playing a key role in food digestion, immune and neural control, antitumor responses, and synthesis of beneficial compounds. It is also regarded to play an important role in obesity and diabetes processes: metabolites and bacterial components of gut microbiota are regarded to the initiation and progression of type 2 diabetes by regulating inflammation, immunity, and metabolism. Many studies have investigated the role of gut microbiota in diabetes, and increasing evidence has demonstrated that fecal microbiota transplantation and probiotic capsules are useful strategies in preventing the disease, indicating gut microbiota as an appropriate therapeutic target in diabetes processes. Indeed, lower microbial biodiversity is a hallmark of these patients [68]. Interestingly, the possible interactions of S100B with microbiota have been recently investigated. First, based on the microbiota composition, proteins putatively interacting with S100B domains were found in silico, both in healthy subjects and inflammatory bowel disease patients, in a reduced number in the latter samples, also exhibiting differences in interacting domain occurrences between the two groups. These results offered the conceptual framework to investigate the role of S100B as a candidate signaling molecule in the microbiota/gut communication machinery [69]. These in silico inferences were experimentally confirmed in mice, where S100B levels correlated with microbiota biodiversity (Shannon values), and the correlation was significantly reduced after treatment with the S100B inhibitor pentamidine [70], indicating that the correlation was influenced by the modulation of S100B activity. Thus, the protein is a constituent of enteroglial cells [71], which correspond to astrocytes in the enteric nervous system. S100B reasonably is also released by these cells and might mediate the regulation of the intestinal microbiota, potentially participating in microbiota-dependent processes. It may be also relevant, in this respect, that S100B may also be taken with food, being a natural constituent of mammalian milk [72,73]. Of course, the above information at present does not suggest any therapy based on S100B through food.

Finally, the exploration of a possible role in obesity/diabetes processes of a protein, such as the S100B, concentrated both in the nervous and adipose tissue, induces a consideration of the possible intriguing relationships linking diabetes to a devastating and still largely unknown pathological condition of the central nervous system, such as Alzheimer’s disease (AD), which has been identified as the most common type of dementia, being the sixth leading cause of death in the United States and the fifth leading cause of mortality in people 65 and older [74]. First, higher rates of dementia have been reported among subjects with diabetes [75,76]. Data have also been reported indicating that insulin resistance, which is known to be a hallmark of the most diffuse form of diabetes (type 2), as above indicated, is also frequently associated with AD dementia, which thus has also sometimes been regarded as type 3 diabetes, essentially related to peculiarities of glucose metabolism within the brain [77,78,79]. In this respect, the observed relationship linking S100B levels to insulin resistance in psychiatric patients may be intriguing [54], although experimental data correlating S100B to the putative connection between diabetes and AD dementia at present are lacking, thus deserving further investigation.

## 5. Conclusions

In conclusion, taken together, the above data suggest a role for adipose S100B in obesity/diabetes processes and for non-neurological metabolic diseases in general terms (Figure 2), re-proposing the putative role played by astrocytic S100B in neuroinflammatory/neurodegenerative processes. Necessarily, additional in vivo studies will be needed to offer a solid basis for this possibility. Hopefully, this review will constitute a stimulation towards studies addressing the putative role of S100B protein in obesity/diabetes processes. In any case, once more, S100B appears to be putatively at the crossroads of different pathological conditions, even involving different tissues and body districts. As a consequence, the protein might even be proposed as a putative multifaceted therapeutic target for different disorders displaying different origins and symptoms, but sharing pathogenic processes involving S100B, reasonably attributable to inflammatory processes.

## Figures and Tables

**Figure 1 ijms-25-00776-f001:**
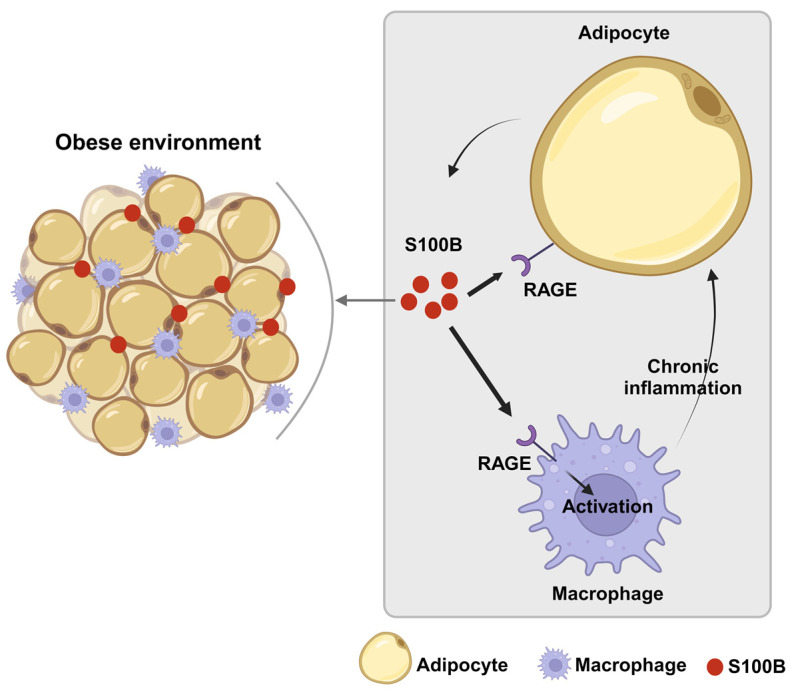
Schematic representation of the paracrine loop involving S100B between adipocytes and macrophages in the development of fatty tissue dysfunction. During obesity processes, adipocytes might release S100B, which acts in a paracrine manner, as an adipokine. In a RAGE-mediated manner, S100B upregulates in macrophages inflammatory cytokines, which, in turn, can stimulate S100B release from adipocytes thus sustaining inflammation via macrophage stimulation.

**Figure 2 ijms-25-00776-f002:**
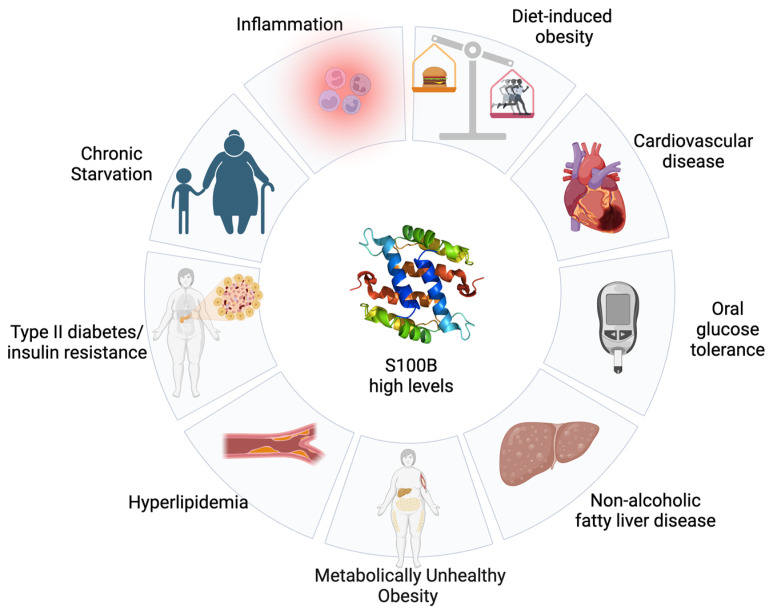
High levels of S100B are associated with several non-neurological metabolic disorders.

**Table 1 ijms-25-00776-t001:** S100B modulation in adipose tissue in different experimental models.

Experimental Model	Treatment	S100B Behavior	Reference
Rat epidydimal fat pads or Isolated adipocytes	Administration of epinephrine or ACTH	Release	Suzuki et al., 1984 [42]
Rat epidydimal adipose tissue	Cathecolamine injection	Reduced content	Suzuki et al., 1984 [44]
Rat epidydimal fat pads	Administration of insulin induced by epinephrine, ACTH, or isoproterenol	reduction in release	Suzuki and Kato 1985 [45]
Epidydimal fat pads from diabetic or starved rats	Untreated	Increased release and reduced content	Suzuki and Kato 1985 [45]

## Data Availability

Not applicable.

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
