# Peer review of "The Multifaceted S100B Protein: A Role in Obesity and Diabetes?"

_ijms, 2024, doi:10.3390/ijms25020776_

Round 1
Reviewer 1 Report
Comments and Suggestions for Authors
In the review manuscript titled " The multifaceted S100B protein: a role in obesity and diabetes?”, Michetti et al. reviewed the potential roles of S100B in the context of obesity and diabetes. Below are my comments and suggestions.
The language in this manuscript is clear, concise, and appropriate for a scientific audience. The review provides a comprehensive background on the S100B protein, its abundance in the nervous system and presence in other tissues, including adipose tissue. The authors effectively set up the premise of investigating the relationship between S100B and obesity-related metabolic disorders. The discussion effectively ties together the multifaceted roles of S100B, emphasizing its potential impact in obesity and diabetes. The manuscript does an excellent job of highlighting the gap in current research regarding the non-neural roles of S100B and proposes intriguing future research directions. The conclusions drawn are well-supported by the literature reviewed. The authors successfully reiterate the potential significance of S100B in obesity and diabetes, advocating for further research in this area.
Minor suggestion 1: while the manuscript contains comprehensive textual information, the inclusion of figures or tables summarizing the roles of S100B in different tissues or its interactions with other biomolecules could enhance understanding.
Minor suggestion 2: include a more detailed figure summarizing the interplay between S100B and metabolic disease progression.
In summary, the manuscript presents a well-researched, insightful analysis of the S100B protein's roles in obesity and diabetes, making a significant contribution to the field. It is recommended for publication following minor enhancements for clarity and comprehension.
Author Response
Responses to reviewers
Reviewer 1
The language in this manuscript is clear, concise, and appropriate for a scientific audience. The review provides a comprehensive background on the S100B protein, its abundance in the nervous system and presence in other tissues, including adipose tissue. The authors effectively set up the premise of investigating the relationship between S100B and obesity-related metabolic disorders. The discussion effectively ties together the multifaceted roles of S100B, emphasizing its potential impact in obesity and diabetes. The manuscript does an excellent job of highlighting the gap in current research regarding the non-neural roles of S100B and proposes intriguing future research directions. The conclusions drawn are well-supported by the literature reviewed. The authors successfully reiterate the potential significance of S100B in obesity and diabetes, advocating for further research in this area.
Minor suggestion 1: while the manuscript contains comprehensive textual information, the inclusion of figures or tables summarizing the roles of S100B in different tissues or its interactions with other biomolecules could enhance understanding.
Minor suggestion 2: include a more detailed figure summarizing the interplay between S100B and metabolic disease progression.
In summary, the manuscript presents a well-researched, insightful analysis of the S100B protein's roles in obesity and diabetes, making a significant contribution to the field. It is recommended for publication following minor enhancements for clarity and comprehension.
We want to thank the Reviewer for kind comments and useful suggestions. In this respect, we build an additional new figure to implement and enhance the understanding of the manuscript, as suggested by the Reviewer.
Reviewer 2 Report
Comments and Suggestions for Authors
This review article by Michettii et al describes the possible role of S100B protein on obesity and diabetes. It is an interesting topic and considering such a high prevalence of obesity and diabetes, it is important to understand their pathology and possible therapeutic options. S100B is well studied in the brain and authors try to gather evidence/literature showing that this may be related to obesity and diabetes. However, the some of evidence from different manuscript is logically connected and many of the evidence are based on tissue culture studies. For obesity and diabetes, it is quite important to provide articles based on the in vivo study. It is a bit premature to conclude the in vivo outcomes simply based on the tissue culture model. Moreover," section 4. S100 protein in obesity/diabetes and related conditions", all the literature simply shows the correlation without any causation. Hence its is not right to conclude that S100B contributes to low-grade inflammation. Reference 66 was provided as evidence for S100B and diabetes through beta cell. Since STZ is simply kills off beta cells chemically (hence the development of diabetes or hyperglycemia itself is artificial system.) Therefore, a lack of beta cell death in S100B KO mice under STZ can be interpreted as "resistance to diabetes." The section for microbiota, also has the premature conclusion that S100B can be treatment through food just because 1) S100B changed microbiome diversity and 2)there are many microbiome-dependent processes. Finally, the figure 1 does not add any value. Authors have emphasized through the article that RAGE is the receptor for S100B and those are only things shown in the figure.
Comments on the Quality of English LanguageThis manuscript may need moderate editing from a native English writer.
